# Developing and validating the Japanese version of the Referential Thinking Scale: A cross-sectional study

Jun Sasaki[1]*, Seiji Muranaka[1], Kotomi Arahata[2], Atsushi Sato[3]

**1** Graduate School of Human Sciences, Osaka University, Suita, Osaka, Japan, **2** Toyama Prefectural Mental Health Center, Toyama, Toyama, Japan, **3** Graduate School of Humanities, Arts, and Social Sciences, University of Toyama, Toyama, Toyama, Japan

* sasaki.jun.hus@osaka-u.ac.jp

## Abstract

It has been shown that ideas of reference in the context of paranoia (IoR-P) and schizophrenia spectrum disorders (IoR-S) are caused by different psychological constructs. Although it is well known that both IoR-P and IoR-S are frequently evoked during the same period of life, how they interact with each other is unknown. The purpose of the present study was to develop the Japanese version of the Referential Thinking Scale (J-REF) to assess IoR-S, examine its validity and reliability, and explore the predictors of IoR-P and IoR-S. In this study, several subgroups of Japanese individuals in their 20s were included in the analysis. The J-REF had high internal consistency, high test-retest reliability, good convergent, and discriminant validity. Two hierarchical regression analyses showed that public self-consciousness predicted the manifestation of IoR-P, while the dimensions of schizotypy predicted that of IoR-S. Moreover, social anxiety and negative moods could cause IoR-P and IoR-S. This study directly showed the existence of two different types of ideas of reference in terms of their predictors. It is also significant in that it first examined referential thinking using the REF scale in the context of Asia and showed that there may not be much difference in the frequency of ideas of reference from other cultures. Future research directions are also discussed.

## Introduction

Ideas of reference (IoR) is a frequently observed psychological symptom in a non-clinical sample [1]. IoR is characterized by a self-referential process in which neutral external events or the actions of others are believed to be directed toward oneself. A classic way of understanding IoR traces its origin to Ernst Kretschmer, who suggests that IoR is mainly transient and commonly occurs during the developmental stages of adolescence and young adulthood, although there are some shifts to paranoia. On the other hand, Freeman et al. [1] showed that about 60% of the non-clinical population (average age, 23.0 years (standard deviation; SD = 6.1, range = 17–61)) experienced an IoR ("*People are laughing at me*"). Fenigstein and Vanable [2] also found that almost all university students falsely believed that others' actions were directed

---

**Data Availability Statement:** Our relevant data is available at Open Science Framework (https://osf.io/p6vzf/).

**Funding:** This research was supported by a Grant-in-Aid (21730548) for Young Scientists from Japan

Society for the Promotion of Science. The funders had no role in study design, data collection and analysis, decision to publish, or preparation of the manuscript.

**Competing interests:** SM received personal fees from a for-profit company CureApp Inc. All other authors have no competing interests. This does not alter our adherence to PLOS ONE policies on sharing data and materials.

at them. Thus, IoR is experienced by many people not only during adolescence and young adulthood.

Empirical research on IoR has mainly been conducted in the context of paranoia and schizophrenia spectrum disorders. According to Freeman et al. [1], IoR is classified as a form of paranoia. Freeman et al. 's [1] cognitive hierarchy model of paranoia posits some levels of suspiciousness, depending on the severity of the perceived threat. Persecutory beliefs are classified in the upper paranoia hierarchy. The perceived threat is stronger, more distressed, and accompanied by conviction. A relatively small number of participants experience this level. On the other hand, IoR and social anxiety are categorized as lower levels of hierarchy. Distress and perceived threat at this level are relatively small. The theme of this level involves interpersonal concerns, which are common in the general population. At the lower level, social anxiety and paranoia are thought to overlap; that is, IoR is thought to be built on social evaluative concerns. Similarly, Tone, Goulding, and Compton [3] found that subclinical paranoia was positively correlated with social anxiety in a college sample. Stefanis et al. [4] showed that IoR and social anxiety load on a paranoia factor, using confirmatory factor analysis. Thus, in the context of paranoia, IoR is thought to have a similar sensitivity to social anxiety.

Previous studies suggest that public self-consciousness is involved in the manifestation of IoR in the context of paranoia (IoR-P). For example, Fenigstein and Vanable [2] showed that public self-consciousness was positively correlated with a paranoid tendency, that is, "misperception of other's behaviors as being directed or targeted toward the self," in a university sample. The Self-Reference Scale (SRS; Kaneko [5]) focuses more on self-referential aspects, while many items of the Paranoid Scale [2] address more depressive themes than persecutory ones [6]. Indeed, Kaneko [5] found that SRS was positively correlated with public self-consciousness and social anxiety but not with private self-consciousness. Moreover, Combs and Penn [7] showed that subclinical paranoia was associated with greater self-consciousness.

However, the manifestation of IoR has been associated with schizophrenia spectrum disorders. According to the Diagnostic and Statistical Manual of Mental Disorders, 5th Edition (DSM-5 [8]), IoR is defined as "the feeling that causal incidents and external events have a particular and unusual meaning that is specific to the person. An idea of reference is to be distinguished from a delusion of reference, in which there is a belief that is held with delusional conviction" (p. 823). IoR is closely related to psychosis, with statistics showing its prevalence in more than half of schizophrenia patients [9]. IoR is also known to be predictive of schizophrenia and other psychotic disorders [10], and is recognized as a prodromal sign [11] and relapse signal [12].

IoR in the context of schizophrenia spectrum disorders (IoR-S) is thought to be caused by a slight form of thought disorder [13]. "Cognitive slippage" [14], considered the mildest type of such thought disorder, is characterized by pathological associations and a difficulty in keeping track of one's thoughts [15]. Cognitive slippage should be detected more frequently in schizotypal people than in healthy controls [16]. Rado [17] first proposed the concept of "schizotypy" as a phenotype of inheritance common to schizophrenia and schizotypal personality. According to Meehl's model [14], certain genes influence brain developmental processes by encoding specific "synaptic control system abnormalities" in the central nervous system. Genetic factors and social learning schedules interact to form a personality structure called schizotypy, and various schizotypal disorders emerge when individuals encounter stressors. Schizotypy is assumed to be the genetic background of IoR-S. More recently, self-disorders [18] especially contributes to the onset of Schizophrenia [19, 20]. For example, Examination of Anomalous Self-Experience scale covers IoR along with this kind of experiences [21]. In this meaning, IoR could indicate a potential risk of developing psychotic disorders.

Schizotypy is a multidimensional construct comprising positive schizotypy, negative schizotypy, and disorganization. Positive schizotypy is characterized by unusual cognitive and perceptual experiences, such as odd beliefs, magical thinking, and IoR. Negative schizotypy is characterized by a decreased interest in relationships and flattened affect, such as social withdrawal and anhedonia. Disorganization is characterized by odd speech and behavior, reflecting deficits in the organization and expression of thoughts and behaviors. Raine et al. [22] showed that a model with these three dimensions showed the best fit to the data and that IoR-S loaded onto a positive schizotypy factor. Cicero and Kerns [23] demonstrated that IoR-S is a distinct facet of positive schizotypy.

Lenzenweger et al. [13] developed the Referential Thinking Scale (REF), a self-report scale to assess IoR-S. They aimed to assess pathological IoR as a schizotypic phenomenon in normal samples and showed that there was no correlation between REF and normative self-consciousness. REF is more comprehensive than the items or subscales of IoR from the Personality Disorder Examination [24] and Schizotypal Personality Questionnaire (SPQ) [25]. Several studies have examined the validity and reliability of the REF. Meyer and Lenzenweger [26] reported that REF for the high schizotypy group was significantly higher than that for the high social anxiety and normal control groups, and the significant difference remained after covarying negative affect. They emphasized that IoR-S does not reflect the influence of negative affect on the interpretation of ambiguous situations (i.e., social anxiety), and that a high level of IoR is specific to schizotypy.

However, there are mixed results in literature regarding the prediction of IoR manifestation. For example, contrary to Lenzenweger et al. [13], Rodríguez-Testal et al. [10] showed that REF scores were significantly and positively correlated with public self-consciousness in healthy youth, healthy adults, and patient groups diagnosed with schizophrenia and other psychotic disorders, anxiety disorders, and depressive disorders, ($r$ = .342, .274, and .225, ps < .001, respectively). Senín-Calderón et al. [27] also showed a significant positive correlation between public self-consciousness and REF in a group randomly and equally selected from normal controls and patients diagnosed with schizophrenia, mainly with paranoid and delusion disorders, mood disorders, anxiety disorders, and so on ($r$ = .347, $p$ < .01). They further revealed that public self-consciousness fully mediated the relationship between anxiety and referential thinking in both the patient group and control groups. These results highlight the possibility that public self-consciousness involves the manifestation of IoR-S along with positive schizotypy. However, the possibility that positive schizotypy is involved in the manifestation of IoR-P remains unclear. Indeed, schizotypy was continuously distributed in the normal sample [28] and contributed to the emergence of psychotic-like symptoms in non-screened university students [29], while high-schizotypy individuals, who are at higher risk to develop schizophrenia, comprised about 10% of the population [30]. The psychological mechanism of manifestation can be different in the two types of IoR, which could cause a mismatch of potential treatment strategies. It is clinically important to recognize the differences between the two types of IoR. However, no study has simultaneously examined whether public self-consciousness and positive schizotypy moderate the manifestation of these two types of IoR.

To investigate these questions, the present cross-sectional study aimed to develop the Japanese version of the REF (J-REF) and examine its convergent validity, discriminant validity, and reliability. Based on prior research that developed and validated the REF [13, 26], we hypothesized as follows:

J-REF would be more strongly and positively correlated with IoR and positive schizotypy than with negative schizotypy and disorganization.

J-REF would have a weaker association with self-consciousness, mood, depression, and anxiety than with schizotypy dimensions.

Although people of all ages experience IoR, this study examined the reliability and validity of the J-REF scale in a sample of people in their 20s, the same age group as the research that developed the REF scale [13].

The second aim was to investigate the predictors of the manifestation of IoR-P and IoR-S, exploratorily. As mentioned previously, there are mixed results in the literature regarding the prediction of IoR manifestation [e.g., 10, 13, 27]. This might have been caused by a covariate factor that was uncontrolled in analyzing the relationship between IoR and other constructs. In fact, Brown et al. [31] highlighted the methodological difficulty in addressing the relationship between psychopathologies that share some points with each other. In this study, the Social Phobia Scale (SPS) [32] was used as one of covariates, considering that the nature of IoR-P and IoR-S share the feature of misunderstanding that one is being observed by others. As previously stated, Stefanis et al. [4] demonstrated a relationship between IoR-P and social anxiety. Morrison and Cohen [33] mentioned that SPS is more appropriate in assessing the fear of being observed by others, while the Social Interaction Anxiety Scale [32] represents the fear of negative evaluation in the situation of being observed by others—Would public self-consciousness solely predict the manifestation of IoR-P? Alternatively, would positive schizotypy solely predict the manifestation of IoR-S? Furthermore, if the interaction effects were significant, public self-consciousness or positive schizotypy would moderate the manifestation of IoR. These findings would enrich our knowledge of the etiology of IoR.

## Material and methods

### Participants and procedure

A total of 300 participants (150 females and 150 males) aged 20–29 years ($M_{age}$ = 25.57 years, $SD$ = 2.70) were selected through an online survey company (Cross-Marketing, Inc.). Tinsley and Tinsley [34] recommended a ratio of five to ten participants per item for factor analysis. As the REF has 34 items, 170–340 participants would suffice. In addition to budgetary restrictions, the sample size for the survey was determined. After four weeks, a random subgroup of 150 participants (75 females and 75 males; $M_{age}$ = 25.41 years, $SD$ = 2.78) responded to the J-REF again to assess test-retest reliability. All participants were recruited during March to April 2022.

The inclusion criteria were as follows: (a) residence in Japan; (b) people in their 20's, who were of similar age to individuals enrolled in the study by Lenzenweger et al. [13]; (c) gender such that an equal number of males and females was obtained; (d) those who correctly responded to an instructional manipulation check [35]; and (e) those who correctly responded to an age consistency check.

### Measures

**Japanese version of Referential Thinking Scale (J-REF).** The J-REF was developed using the following steps: first, permission was obtained from Dr. Lenzenweger to translate the original version of the REF [13] into Japanese. We asked a translation service company to translate the original version to Japanese. Two Japanese clinical psychologists and one psychiatrist specializing in psychotic disorders reviewed and modified the translations so that each item corresponded well with the original items. After that, we asked another translation service company to translate the J-REF back to English. After the author of the original REF checked the back-translated version, he confirmed the equivalency of the 15 items and commented on 19 items for correction. The main comment suggested that the items in the Japanese version be translated into a more neutral experience. In response, we explained that although different expressions were used in English, they were the same when translated into Japanese, and that faithful

translation into English would make Japanese unnatural. Consequently, four items were approved. We modified the remaining 15 items and asked other translation service companies to back-translate them. Finally, the J-REF was finalized after the original authors confirmed that the second back-translation had the same meaning as the original version.

Lenzenweger et al. [13] recommend adding the number of items × 4 = 132 filler items to prevent the intention of REF items from being detected; however, at the same time, the problem of careless responding (e.g., [36]) is undeniable. This study included 20 filler items, the same percentage of items (approximately 60%) as Kaneko's study [5]. The filler items in the present study were developed based on the validity scales of the Minnesota Multiphasic Personality Inventory-2 (F scales). Thus, participants were asked to respond to 54 items, including 34 J-REF and 20 filler items, in a true-false style. The original version of the REF has five subscales: "Laughing, Commenting," "Attention, Appearance," "Guilt, Shame," "Songs, Newspapers, Books," and "Reactions."

The original version of REF has high internal reliability (α = .85) and high four-week test-retest reliability (r = .86) for the total REF score. They also reported the high construct validity, showing stronger correlation of REF with the schizotypal constructs, such as perceptual aberration (r = .53) and magical ideation (r = .61) and no significant correlation with self-consciousness (r = -.10) [13].

**Self-Reference Scale (SRS).**   The IoR-P was measured using the SRS [5]. This scale comprises 12 items (e.g., "When I see my friends talking in private, I worry that they are talking bad about me") using a five-point Likert scale (1 = *strongly disagree* to 5 = *strongly agree*). As in Kaneko's study [5], seven filler items made from the validity scales of the Minnesota Multiphasic Personality Inventory-2 (F scales) were included. Cronbach's α coefficient was .90. The three-week test–retest reliability was .76. The SRS has good construct validity, with convergent and discriminant measures [5].

**Schizotypal Personality Questionnaire (SPQ).**   Schizotypal personality was measured using the Japanese version of the SPQ [25, 37]. This scale comprises 74 items with nine subscales: (a) ideas of reference (9 items), (b) odd beliefs or magical thinking (7 items), (c) unusual perceptual experiences (9 items), (d) suspiciousness (8 items), (e) excessive social anxiety (8 items), (f) no close friends (9 items), (g) constricted affect (8 items), (h) odd or eccentric behavior (7 items), and (i) odd speech (9 items). Participants responded to each item using the true-false style. The cognitive–perceptual (SPQ-CogPer) dimension (i.e., positive schizotypy) comprises (a), (b), (c), and (d). The interpersonal (SPQ-Inter) dimension (i.e., negative schizotypy) comprises (d), (e), (f), and (g). The disorganized (SPQ-Disorg) dimension (i.e., disorganization) comprises (h) and (i). The SPQ displays good internal consistency (α = .63–.86) and test-retest reliability (r = .76–.86). The SPQ has good construct validity with convergent measures [37].

**Self-Consciousness Scale.**   Self-consciousness was measured using the Japanese version of the self-consciousness scale [38, 39]. This scale comprises 21 items with two subscales: (a) public self-consciousness (SC-PUB; 11 items) and (b) private self-consciousness (SC-PRI; 10 items). Participants responded to each item using a seven-point Likert scale (1 = *strongly disagree* to 7 = *strongly agree*). The self-consciousness scale displays good internal consistency (r = .78, .75, respectively) and has good construct validity with convergent measures [39].

**State-Trait Anxiety Inventory (STAI-S).**   Individual differences in state anxiety were measured using the Japanese version of the State-Trait Anxiety Inventory (STAI) [40, 41]. This scale comprises 20 items rated on a four-point Likert scale (1 = *not at all* to 4 = *very much so*). The STAI-S displays good internal consistency (α = .92) and construct validity [41].

**Self-Rating Depression Scale (SDS).**   The frequency of depressive symptoms over the past week was measured using the Japanese version of the Self-Rating Depression Scale (SDS)

[42, 43]. This scale comprises 20 items rated on a four-point Likert scale (1 = *a little of the time* to 4 = *most of the time*). The SDS displays good internal consistency ($\alpha$ = .63–.86) and test-retest reliability ($r$ = .76–.86). It has good construct validity with convergent measures [43].

**Positive Affect and Negative Affect Scale (PANAS).**   Current affect was measured using the Japanese version of the Positive and Negative Affect Scale (PANAS) [44, 45]. This scale comprises 16 items with two subscales: (a) positive affect (PA; 8 items) and (b) negative affect (NA; 8 items). Participants responded to each item using a six-point Likert scale (1 = *not at all* to 6 = *very much*). PANAS displays good internal consistency ($\alpha$ = .83, .82, respectively) and construct validity [45].

**Social Phobia Scale (SPS).**   Scrutiny fear was measured using the Japanese version of the Social Phobia Scale (SPS) [32, 46]. This scale comprises 20 items using a five-point Likert scale (0: *not at all characteristic or true of me* to 4: *extremely characteristic or true of me*). The SPS displays good internal consistency ($\alpha$ = .91) and has good construct validity, with convergent and discriminant measures [46].

## Statistical analyses

First, an exploratory factor analysis (EFA) with a robust weighted least squares (WLS) estimator and oblimin rotation was conducted based on the tetra-choric correlations among the 34 items of J-REF to determine its latent structure. The obtained structure was subjected to confirmatory factor analysis (CFA) using structural equation modeling (SEM). The robust diagonally weighted least squares estimation method (DWLS) was utilized using tetrachoric correlations to conduct CFA. We compared the model obtained in the EFA with potential models (null model, unidimensional model, original model of Lenzenweger et al. [13]; and alternative model of Rodríguez-Testal et al. [10]). The goodness-of-fit of the model was evaluated using chi-square ($\chi^2$), goodness-of-fit index (GFI), adjusted goodness-of-fit index (AGFI), comparative fit index (CFI), non-normed fit index (NNFI), incremental fit index (IFI), root mean square error of approximation (RMSEA), and a 90% confidence interval (CI). The model fits well when CMIN/DF < 2 [47], GFI > .95, AGFI > .90, CFI > 0.95, NNFI > 0.95, IFI > 0.95, and RMSEA < 0.05 [48]. Second, the internal consistency of the J-REF was assessed using Cronbach's alpha. The test-retest reliability of the J-REF was assessed using the intraclass correlation coefficient (ICC). Third, convergent validity was examined by assessing Pearson's correlation coefficients. Tests of differences in correlation coefficients between the J-REF and hypothesized variables were conducted to confirm discriminant validity. Finally, two hierarchical regression analyses were conducted to examine the influence of the related variables on J-REF and SRS. Sociodemographic variables (i.e., sex and age) were entered in step 1, followed by SRS/J-REF (step 2), positive and negative affect (step 3), social anxiety (step 4), public and private self-consciousness (step 5), schizotypal traits of positive and negative schizotypy, and disorganization (step 6). The significance level was set as 0.05.

All analyses were conducted using the R software (ver. 4.2.1). The psych package (ver. 2.2.5) was used for factor analysis and to calculate the polychoric correlation and reliability coefficients of the scales. The lavaan package (ver. 0.6–12) was used to conduct CFA using SEM.

## Ethical considerations

All procedures were performed in accordance with the Declaration of Helsinki and approved by the Ethics Committees of the Graduate School of Human Sciences at Osaka University (approval ref no. 21–111). Participation in this study was voluntary, and electronic consent was obtained from all participants by clicking on the "Agree" button that was mandatory to proceed to the questionnaire. The cover page provided the following relevant information:

participation in the survey was voluntary, participants could stop answering the questionnaire if they felt uncomfortable, their anonymity would be protected, the information obtained would not be used for any purpose other than the research, and all answers were analyzed statistically. Participants were provided with prescribed points for completing all sets of the questionnaires.

## Results

### Factor structure of J-REF

Bartlett's test of sphericity [$\chi^2(561)$ = 7855.90, p < .001] and a Kaiser–Meyer–Olkin (KMO) value of 0.68 showed that our 34 items of the J-REF were suitable for factor analysis [49]. The observed eigenvalues indicated a six-factor structure with $\lambda$ > 1.0 (14.61, 2.37, 1.73, 1.37, 1.29, 1.08, 0.98). In contrast, parallel analysis and minimum partial average (MAP) analysis proposed 18- and two-factor solutions, respectively. Considering factor analyses in prior research indicating the multidimensionality of the REF scale [10, 13], a five-factor solution was first selected. Exploratory factor analysis with oblimin rotation of the 34 items was performed with a five-factor solution using WLS estimation. However, a Heywood case occurred, which would be caused by the sample size used for EFA (e.g., [50]). The online survey company collected additional data in the first survey as a subsample aside from the 300 participants, to ensure the confirmation of test-retest reliability using the sample that met the inclusion criteria. Another 300 participants (150 females and 150 males) were randomly selected and added from the subsample. A total of 600 participants (300 females and 300 males) aged 20–29 years ($M_{age}$ = 25.64 years, $SD$ = 2.65 years) were used for EFA and CFA. To avoid type-I error rate inflation, we used the original sample (n = 300) for the evaluation of reliability and validity and hierarchical regression analyses.

Descriptive statistics of each item of the J-REF and their gender differences (n = 600) were shown in S1 and S2 Tables. It was found that all items of the J-REF were not normally distributed along with the total score of the J-REF. Scores of males in Item 17, 27, and 32 were higher than that of the females and the score of the females in Item 6 were higher than that of the males, although there was no statistical difference in the total score of J-REF between female and male.

Bartlett's test of sphericity [$\chi^2(561)$ = 6287.51, $p$ < .001] and a KMO value of 0.62 showed that our 34 items of the J-REF were suitable for factor analysis [49]. A rule of thumb indicates that a lower value of KMO is calculated with tetrachoric correlations than Pearson's correlations. The observed eigenvalues indicated a five-factor structure with $\lambda$ > 1.0 (14.98, 2.28, 1.48, 1.06, 1.01, and 0.79). Parallel analysis and minimum partial average (MAP) analysis proposed 16- and two-factor solutions, respectively. Considering the factor analyses in prior research indicating the multidimensionality of the REF scale [10, 13], a five-factor solution was first selected. EFA with oblimin rotation of the 34 items was performed with a five-factor solution using the WLS estimation. Only one item (item 19) was loaded into the fifth and fourth factors in the five- and four-factor solutions, respectively. A three-factor model was then selected as the most appropriate solution (Table 1).

The factor analysis results revealed that the subscales established in the previous study were categorized across several factors. This would mean that each factor of the J-REF shares relatively similar IoR characteristics in the Japanese sample. Indeed, the inter-factor correlations were moderately high (r = .34–.57). Therefore, each factor was named based on its comparative characteristics. Factor 1 consisted of "Laughing, Commenting (10 items; 1, 3, 14, 2, 6, 9, 4, 24, 18, and 7)," "Guilt, Shame (3 items; 31, 33, and 30)," and "Attention, Appearance (3 items; 8, 5, and 23)" derived from Lenzenweger, et al. [13], while Factor 2 consisted of "Attention,

**Table 1. Standardized item-factor loadings for the J-REF scale and inter-factor correlation matrix (n = 600).**

| Item | Factor 1 | Factor 2 | Factor 3 |
|------|----------|----------|----------|
| 1 | .91 | -.16 | -.04 |
| 3 | .90 | -.17 | -.01 |
| 14 | .84 | -.02 | -.01 |
| 2 | .79 | -.04 | .10 |
| 6 | .78 | -.05 | .14 |
| 9 | .70 | -.01 | .24 |
| 4 | .69 | .14 | -.11 |
| 31 | .68 | .21 | -.09 |
| 8 | .65 | .14 | .14 |
| 24 | .63 | .28 | -.01 |
| 33 | .62 | .18 | -.03 |
| 18 | .62 | .19 | .10 |
| 5 | .61 | .21 | .08 |
| 30 | .60 | .19 | -.04 |
| 7 | .58 | .33 | .06 |
| 23 | .33 | .30 | .30 |
| 34 | -.04 | .75 | .11 |
| 20 | -.06 | .75 | .26 |
| 32 | .11 | .66 | -.06 |
| 22 | .19 | .64 | -.04 |
| 28 | .06 | .61 | -.20 |
| 25 | -.18 | .61 | .16 |
| 27 | .29 | .51 | -.03 |
| 17 | .21 | .49 | -.14 |
| 21 | .27 | .49 | .12 |
| 16 | .23 | .45 | -.09 |
| 11 | .25 | .42 | .09 |
| 26 | .10 | .41 | .10 |
| 29 | .30 | .40 | -.11 |
| 15 | -.02 | -.07 | .92 |
| 13 | .16 | .08 | .76 |
| 10 | .16 | .21 | .54 |
| 12 | .06 | .39 | .46 |
| 19 | .03 | -.12 | .17 |
| Factor 1 | | .57 | .34 |
| Factor 2 | | | .35 |

Appearance (4 items; 20, 32, 25, and 26)," "Songs, Newspapers, Books (4 items; 22, 27, 16, and 11)," "Guilt, Shame (3 items; 28, 21, and 29)," and "Reactions (2 items; 34, 17)"; Factor 3 included "Songs, Newspapers, Books (3 items; 15, 13, and 10)," "Attention, Appearance (1 item; 19)," and "Reactions (1 item; 12)." Although Factor 1 and Factor 2 shared the items from "Attention, Appearance (4 items)" and "Guilt, Shame (3 items)," Factor 1 specifically had more items from "Laughing, Commenting" (10 items). Factor 1 was thus named "Laughing, Commenting." Factor 2 consisted of items from four other factors that were not from "Laughing, Commenting." These items represented being mimicked and seen through by others, then named as "Attention, mimicking, seen-through." Factor 3 mainly consisted of and was named "Songs, Newspapers, Books." All the items and instructions for the J-REF are presented in S3 Table.

**Table 2. Fit indices of CFA for the five models (n = 600).**

| Models | $X^2$ | df | GFI | AGFI | CFI | NNFI | IFI | TLI | RMSEA [90% CI] |
|---|---|---|---|---|---|---|---|---|---|
| Model 1 | 5891.241 | 561 | | | | | | | |
| Model 2 | 853.183 | 527 | .972 | .968 | .939 | .982 | .983 | .935 | .032[.028, .036] |
| Model 3 | 735.130 | 524 | .978 | .975 | .960 | .992 | .993 | .958 | .026[.021, .030] |
| Model 4 | 727.949 | 517 | .978 | .975 | .960 | .992 | .993 | .957 | .026[.022, .030] |
| Model 5 | 692.292 | 522 | .980 | .978 | .968 | .996 | .996 | .966 | .023[.018, .028] |

Note: Model 1: Null model; Model 2: Unidimensional model; Model 3: The three-factor model obtained in the EFA; Model 4: Original structure of the REF scale obtained by Lenzenweger, et al. [13]; Model 5: Alternative structure of the REF scale obtained by Rodríguez-Testal et al. [10].

Next, CFAs were conducted to compare the fit of the following potential models to the J-REF data: null model (Model 1), unidimensional model (Model 2), three-factor model obtained in the EFA (Model 3), original structure by Lenzenweger et al. [13] (Model 4), and alternative structure by Rodríguez-Testal et al. [10] (Model 5). The CFA fit indices indicated an excellent fit for all models, as shown in Table 2. In particular, all fit indices for Model 5 were higher than those of the other models (GFI = .980, AGFI = .978, CFI = .968, NNFI = .996, IFI = .996, TLI = .966, RMSEA = .023 [.018, .028]); therefore, we selected this model as our final factor construct of J-REF. Rodríguez-Testal et al.'s model [10] consisted of one second-order factor and five first-order factors as follows: F1: "Attention, Appearance (Items 8, 9, 18, 21, 24, 25, 26, and 32)," F2: "Laughing, Commenting (Items 1–4, 6, and 14)," F3: "Songs, Newspapers, Books (Items 10–13)," F4: "Guilt, Shame (Items 28–31, and 33)," and F5: Causal Explanations (Items 5, 7, 15–17, 19, 20, 22, 23, 27, and 34)". Rodríguez-Testal et al. [10] recommended using the total REF score for analysis. Indeed, in this study, a second-order factor showed high standardized partial regression coefficients for the five first-order factors (F1: .974, F2: .838, F3: .751, F4: .876, and F5: .924). Hence, we used the total J-REF score for the analysis. The results of statistical analysis using each subscale of the J-REF are presented for reference in the following section.

## Internal consistency and test-retest reliability

Cronbach's alpha was calculated to examine the internal consistency of the J-REF. Cronbach's α was 0.89, indicating good internal consistency for the J-REF. Cronbach's α for each subscale was 0.71 ("Attention, Appearance"), 0.81 ("Laughing, Commenting"), 0.41 ("Songs, Newspapers, Books"), 0.70 ("Guilt, Shame"), and 0.69 ("Causal Explanations").

The ICC [2, 1] was calculated to examine the test–retest reliability of the J-REF, using data from a random subgroup (n = 150) that responded to the J-REF again after four weeks, as mentioned above. The ICC [2, 1] was 0.79 (95% CI: 0.72 to 0.84), indicating moderate-to-good test–retest reliability of the J-REF for four-week intervals. The ICC [2, 1] for each subscale was 0.74 [0.65, 0.80] ("Attention, Appearance"), 0.78 [0.71, 0.84] ("Laughing, Commenting"), and 0.69 [0.60, 0.77] ("Causal Explanations"), indicating moderate-to-good test-retest reliability; 0.66 [0.56, 0.74] ("Guilt, Shame") and 0.59 [0.47, 0.68] ("Songs, Newspapers, Books"), indicating moderate and poor-to-moderate test-retest reliability, respectively.

## Convergent and discriminant validities

The descriptive statistics and correlations with the total J-REF are shown in Tables 3 and 4. As shown in Table 4, J-REF was positively correlated with many of the other variables, such as SPQ-CogPer ($r = .75$, $p < .001$), SPQ-Disorg ($r = .67$, $p < .001$), and SPQ-Inter ($r = .58$, $p <$

**Table 3. Descriptive statistics of the J-REF and other scales.**

|              | N   | Means | SD    | Min. | Max. | Skewness | Kurtosis | α   |
|--------------|-----|-------|-------|------|------|----------|----------|-----|
| J-REF        | 300 | 4.36  | 5.02  | 0    | 29   | 2.00     | 4.72     | .89 |
| SRS          | 300 | 33.49 | 12.20 | 12   | 60   | 0.12     | -0.69    | .95 |
| SC-PUB       | 300 | 49.14 | 12.09 | 13   | 77   | -0.20    | 0.27     | .90 |
| SC-PRI       | 300 | 44.08 | 10.06 | 13   | 70   | -0.06    | 0.39     | .88 |
| SPQ-CogPer   | 300 | 7.49  | 6.53  | 0    | 33   | 1.17     | 1.50     | .91 |
| SPQ-Inter    | 300 | 13.59 | 8.33  | 0    | 33   | 0.20     | -0.90    | .93 |
| SPQ-Disorg   | 300 | 5.31  | 4.22  | 0    | 16   | 0.65     | -0.41    | .87 |
| SDS          | 300 | 47.42 | 9.52  | 21   | 77   | 0.18     | 0.09     | .85 |
| STAI-S       | 300 | 47.18 | 11.74 | 20   | 80   | 0.23     | 0.03     | .93 |
| NA           | 300 | 22.82 | 8.18  | 8    | 48   | 0.39     | 0.30     | .91 |
| PA           | 300 | 21.43 | 7.17  | 8    | 43   | 0.15     | -0.12    | .89 |
| SPS          | 300 | 19.20 | 17.85 | 0    | 80   | 1.19     | 1.04     | .96 |

Note: N = 300. J-REF = Japanese version of Referential Thinking Scale; SC-PUB = Public Self-Consciousness Scale; SC-PRI = Private Self-Consciousness Scale;

SRS = Self-Reference Scale; SPQ-CogPer = positive schizotypy; SPQ-Inter = negative schizotypy; SPQ-Disorg = disorganization; SDS = Self-rating Depression Scale;

STAI-S = State-Trait Anxiety Inventory (A-State); NA = Negative Affect Scale; PA = Positive Affect Scale; SPS = Social Phobia Scale.

.001), with the most positive and strong correlation with SPQ-IoR ($r =. 76, p < .001$). To determine whether J-REF was significantly more correlated with positive schizotypy than negative schizotypy and disorganization, we used the methods by Meng, Rosenthal, and Rubin [51]. These analyses indicated that J-REF was significantly more correlated with SPQ-CogPer than SPQ-Disorg ($Z = 5.32, p < .001$), and SPQ-Inter ($Z = 8.60, p < .001$). Furthermore, there were significant differences in the correlation coefficient of J-REF between SPQ-CogPer and NA ($Z = 7.76, p < .001$), which showed the strongest correlation with J-REF among the schizotypy dimensions. The SDS and STAI-S scores were positively and moderately correlated ($r = .50, .46, p < .001$, respectively). The SC-PUB and SC-PRI were positively and mildly correlated ($r =. 37$, and $.25, p < .001$, respectively). PA was negatively and weakly correlated ($r = −.16, p < .01$).

It was confirmed that there was a relatively stronger correlation between J-REF and SRS ($r = 0.60, p < .001$) than that from different combinations of other variables. To investigate how differently J-REF and SRS were associated with other psychological constructs, we performed partial correlation analyses adjusting for J-REF and SRS, in order.

As shown in Table 5, when adjusting for SRS, the partial correlations of J-REF with SC-PUB and SC-PRI became insignificant ($pr = .01, .08, ns$, respectively), although J-REF correlated significantly and positively with SC-PUB ($r = .37, p < .001$) and SC-PRI ($r = .25, p < .001$) when not adjusting for SRS. In contrast, when adjusting for J-REF, the partial correlations of SRS with SC-PUB and SC-PRI remained significant ($pr = .52, .21, p < .001$, respectively).

When adjusting for SRS, the partial correlations of J-REF with SPQ-CogPer, SPQ-Inter, and SPQ-Disorg remained significant ($pr = .64, .39$, and $.57, ps < .001$, respectively). In contrast, when adjusting for the J-REF, the partial correlations of SRS with SPQ-CogPer and SPQ-Inter ($pr = .17, p < .01; .27, p < .001$) became drastically weaker, but remained significant. The partial correlations of SRS with SPQ-Disorg became insignificant ($pr = .08, ns$).

When adjusting for SRS, the partial correlations of J-REF with SDS, STAI-S, NA, and SPS became weaker, but remained significant ($pr = .32, .27, .28$, and $.33, ps < .001$, respectively); on the contrary, that with PA became insignificant ($pr = −.02, ns$). When adjusting for J-REF, the partial correlations of SRS with SDS, STAI-S, NA, PA, and SPS became weaker, but remained significant ($pr = .23, .24, .31, ps < .001$, respectively; $−.18, p < .01$; and $.41, p < .001$).

**Table 4. Correlation of the J-REF and other scales.**

|  | 1 | 2 | 3 | 4 | 5 | 6 | 7 | 8 | 9 | 10 | 11 | 12 |
|---|---|---|---|---|---|---|---|---|---|---|---|---|
| **1 J-REF** |  | .60*** | .37*** | .25*** | .75*** | .58*** | .67*** | .50*** | .46*** | .51*** | -.16** | .58*** |
| **2 SRS** |  |  | .61*** | .31*** | .54*** | .52*** | .45*** | .46*** | .45*** | .52*** | -.24*** | .61*** |
| **3 SC-PUB** |  |  |  | .61*** | .37*** | .28*** | .31*** | .26*** | .25*** | .39*** | -.05 | .42*** |
| **4 SC-PRI** |  |  |  |  | .30*** | .22*** | .25*** | .24*** | .20** | .34*** | -.02 | .33*** |
| **5 SPQ-CogPer** |  |  |  |  |  | .74*** | .76*** | .49*** | .48*** | .50*** | -.18** | .58*** |
| **6 SPQ-Inter** |  |  |  |  |  |  | .74*** | .57*** | .58*** | .51*** | -.40*** | .57*** |
| **7 SPQ-Disorg** |  |  |  |  |  |  |  | .46*** | .48*** | .48*** | -.18** | .48*** |
| **8 SDS** |  |  |  |  |  |  |  |  | .81*** | .61*** | -.51*** | .51*** |
| **9 STAI-S** |  |  |  |  |  |  |  |  |  | .64*** | -.49*** | .52*** |
| **10 NA** |  |  |  |  |  |  |  | *** |  |  | -.09 | .51*** |
| **11 PA** |  |  |  |  |  |  |  |  |  |  |  | -.31*** |
| **12 SPS** |  |  |  |  |  |  |  |  |  |  |  |  |

Note: N = 300. * $p < .05$, ** $p < .01$, *** $p < .001$. J-REF = Japanese version of Referential Thinking Scale; SC-PUB = Public Self-Consciousness Scale; SC-PRI = Private Self-Consciousness Scale; SRS = Self-Reference Scale; SPQ-CogPer = positive schizotypy; SPQ-Inter = negative schizotypy; SPQ-Disorg = disorganization; SDS = Self-rating Depression Scale; STAI-S = State-Trait Anxiety Inventory (A-State); NA = Negative Affect Scale; PA = Positive Affect Scale; SPS = Social Phobia Scale.

There were no differences in the descriptive statistics, correlation matrix, and partial correlation coefficients of variables between the results using n = 300 and n = 600 (S4–S6 Tables).

## Hierarchical multiple regression analysis

Hierarchical multiple regression analyses performed to examine the predictors of the J-REF and SRS scores are shown in Table 6. All predictive variables were centered to reduce problems with collinearity between the main effect and interaction term [52]. Variance inflation factor (VIF) was calculated to evaluate the models for multicollinearity. The VIF ranged from 1.34 to 3.94, and the tolerance, which is reciprocal of VIF ranged from 0.25 to 0.74, thus showed no multicollinearity issues [53].

**Table 5. Partial correlation of scales after controlling for J-REF and SRS.**

|  | J-REF(SRS) | SRS(J-REF) |
|---|---|---|
| **SC-PUB** | .01 | .52*** |
| **SC-PRI** | .08 | .21*** |
| **SPQ-CogPer** | .64*** | .17** |
| **SPQ-Inter** | .39*** | .27*** |
| **SPQ-Disorg** | .57*** | .08 |
| **SDS** | .32*** | .23*** |
| **STAI-S** | .27*** | .24*** |
| **NA** | .28*** | .31*** |
| **PA** | -.02 | -.18** |
| **SPS** | .33*** | .41*** |

Note: N = 300. * p < .05, ** p < .01, *** p < .001. Control variable is in parenthesis. J-REF = Japanese version of Referential Thinking Scale; SC-PUB = Public Self-Consciousness Scale; SC-PRI = Private Self-Consciousness Scale; SRS = Self-Reference Scale; SPQ-CogPer = positive schizotypy; SPQ-Inter = negative schizotypy; SPQ-Disorg = disorganization; SDS = Self-rating Depression Scale; STAI-S = State-Trait Anxiety Inventory (A-State); NA = Negative Affect Scale; PA = Positive Affect Scale; SPS = Social Phobia Scale.

**Table 6. Hierarchical multiple regression analyses predicting J-REF and SRS.**

| | J-REF | | | | SRS | | | |
|---|---|---|---|---|---|---|---|---|
| | β | $R^2$ | $\Delta R^2$ | $\Delta F$ | β | $R^2$ | $\Delta R^2$ | $\Delta F$ |
| **Step 1** | | | | | | | | |
| SRS/J-REF | 0.60*** | 0.36 | | | 0.60*** | 0.36 | | |
| **Step 2** | | | | | | | | |
| SRS/J-REF | 0.46*** | | | | 0.43*** | | | |
| PA | -0.03 | | | | -0.15** | | | |
| NA | 0.27*** | 0.41 | 0.05 | 22.42*** | 0.28*** | 0.44 | 0.08 | 31.42*** |
| **Step 3** | | | | | | | | |
| SRS/J-REF | 0.34*** | | | | 0.31*** | | | |
| PA | 0.02 | | | | -0.08 | | | |
| NA | 0.19*** | | | | 0.19*** | | | |
| SPS | 0.28*** | 0.46 | 0.04 | 36.36*** | 0.31*** | 0.50 | 0.05 | 41.12*** |
| **Step 4** | | | | | | | | |
| SRS/J-REF | 0.36*** | | | | 0.26*** | | | |
| PA | 0.03 | | | | -0.10* | | | |
| NA | 0.19*** | | | | 0.14** | | | |
| SPS | 0.28*** | | | | 0.22*** | | | |
| SC-PUB | -0.05 | | | | 0.45*** | | | |
| SC-PRI | 0.05 | 0.46 | 0.00 | 0.48 | -0.15** | 0.62 | 0.12 | 45.67*** |
| **Step 5** | | | | | | | | |
| SRS/J-REF | 0.24*** | | | | 0.27*** | | | |
| PA | -0.01 | | | | -0.06 | | | |
| NA | 0.08 | | | | 0.12** | | | |
| SPS | 0.12* | | | | 0.20*** | | | |
| SC-PUB | -0.05 | | | | 0.46*** | | | |
| SC-PRI | -0.03 | | | | -0.15** | | | |
| SPQ-CogPer | 0.48*** | | | | 0.00 | | | |
| SPQ-Inter | -0.20** | | | | 0.15* | | | |
| SPQ-Disorg | 0.27*** | 0.66 | 0.20 | 58.40*** | -0.11 | 0.62 | 0.01 | 2.09 |

Note: N = 300. * $p < .05$, ** $p < .01$, *** $p < .001$. J-REF = Japanese version of Referential Thinking Scale; SC-PUB = Public Self-Consciousness Scale; SC-PRI = Private Self-Consciousness Scale; SRS = Self-Reference Scale; SPQ-CogPer = positive schizotypy; SPQ-Inter = negative schizotypy; SPQ-Disorg = disorganization; SDS = Self-rating Depression Scale; STAI-S = State-Trait Anxiety Inventory (A-State); NA = Negative Affect Scale; PA = Positive Affect Scale; SPS = Social Phobia Scale.

Hierarchical multiple regression analyses performed to examine the predictors of J-REF scores are presented in Table 6. Age and sex were included in step 1 as covariates, and the change in $R^2$ was -0.005. The negative value of the coefficient of determination showed that age and gender did not have explanatory power; therefore, we decided not to include this step in the hierarchical regression analyses. Furthermore, there was a strong correlation between the J-REF and SRS (r = 0.60, p < .001). This shows the overlapping nature of IoR-S and IoR-P. To reveal the specific features of IoR-S, the SRS score was included in step 1 as a covariate, and the change in $R^2$ was 0.36. PA and NA scores were included at step 2 as covariates, and the change in $R^2$ was 0.05 (p < .001). SPS score was included at step 3 as a covariate, and the change in $R^2$ was 0.04, (p < .001). SC-PUB and SC-PRI scores were included in step 4 as covariates, and the change in $R^2$ was 0.00, *ns*. SPQ-CogPer, SPQ-Inter, and SPQ-Disorg scores were included at step 5 as covariates, and the change in $R^2$ was 0.20 (p < .001). All the predictive variables explained 66.2% of the variance in the criterion variable ($R^2 = 0.66$). The results of

step 5 showed that SRS ($\beta$ = 0.24, $p$<.001), SPS ($\beta$ = 0.12, $p$<.05), SPQ-CogPer ($\beta$ = 0.48, $p$<.001), SPQ-Inter ($\beta$ = − 0.20, $p$<.01), and SPQ-Disorg ($\beta$ = 0.27, $p$<.001) significantly predicted J-REF scores.

The hierarchical multiple regression analyses performed to examine the predictors of SRS scores are presented in Table 6. To maintain equivalence with the regression model of J-REF, age and gender were not included in step 1 as covariates, although there was no statistical problem in the analysis. Similarly, the J-REF score was included in step 1 as a covariate, and the change in $R^2$ was 0.36. PA and NA scores were included in step 2 as covariates, and the change in $R^2$ was 0.08 ($p < .001$). The SPS score was included in step 3 as a covariate, and the change in $R^2$ was 0.05 ($p < .001$). SC-PUB and SC-PRI scores were included in step 4 as a covariate, and the change in $R^2$ was 0.12 ($p < .001$). SPQ-CogPer, SPQ-Inter, and SPQ-Disorg scores were included in step 5 as covariates, and the change in $R^2$ was 0.01, *ns*. All the predictor variables explained 61.5% of the variance in the criterion variable ($R^2 = 0.62$). Results of step 4 showed that J-REF ($\beta$ = 0.26, $p$<.001), PA ($\beta$ = −0.10, $p$<.05), NA ($\beta$ = 0.14, $p$<.01), SPS ($\beta$ = 0.22, $p$<.001), SC-PUB ($\beta$ = 0.45, $p$<.001), and SC-PRI ($\beta$ = −0.15, $p$<.01), significantly predicted the SRS score.

## Discussion

The purpose of the present study was two-fold. The first purpose was to develop the J-REF and examine its convergent validity, discriminant validity, and reliability. The second purpose was to investigate the predictors of the manifestation of IoR-P and IoR-S by controlling covariates, exploratorily.

### Psychometric properties of J-REF

This study confirmed the reliability, validity, and factor structure of the J-REF. Cronbach's alpha indicated good internal consistency for the J-REF. The ICC indicated that the test–retest reliability of the J-REF was moderate-to-good according to Koo and Li [54]. Therefore, the J-REF had a high reliability. Furthermore, as hypothesized, the J-REF was strongly correlated with the IoR subscale of SPQ, positive schizotypy, and SRS, indicating convergent validity of the J-REF. Moreover, J-REF was more strongly correlated with negative schizotypy and disorganization than with self-consciousness, mood, and anxiety, indicating discriminant validity of the J-REF.

The Rodríguez-Testal et al. [10] model showed the most excellent fit among the five potential models for factor structure of J-REF, with the following subscale: "Attention, Appearance," "Laughing, Commenting," "Songs, Newspapers, Books," "Guilt, Shame," and "Causal Explanations." Rodríguez-Testal et al. [10] recommended using the total score for analysis. REF was mainly used as a total score (e.g., [13, 55–57]), and its cut-off point was also set by a total score [13]. Indeed, the relationship between second-order and first-order factors was quite high in this study ($\beta$ = .751 to .974, ps < .001). This study also recommended using the total score of 34 items in the J-REF, similar to Rodríguez-Testal et al. [10].

When using the subscales of the J-REF, it is necessary to confirm whether the results of the factor analysis in this study can be replicated and whether the internal consistency is sufficiently high. Although there was high internal consistency in total scores ($\alpha$ = .89) and moderate internal consistency in the four subscales ($\alpha$ = .69–.81), the internal reliability of "Songs, Newspapers, Books" was low ($\alpha$ = .41). This may be due to the small number of items (four) and the fact that this is an experience that is seen by only a small percentage of people that the media reports. Furthermore, the true-false format of the J-REF may also have influenced the low alpha coefficient. For example, when the SPQ involved "Constricted affect" and "No close

friends" to be responded to in true-false, the alpha coefficients were .58 and .68, respectively. However, the internal consistency increased to .75 and .83, respectively, on a five-point Likert scale ("strongly disagree," "disagree," "neutral," "agree," and "strongly agree") [58].

## Predictor of IoR-S and IoR-P

A strong significant correlation was confirmed between SRS and J-REF. Partial correlation analyses revealed that the significant correlation of J-REF with public and private self-consciousness diminished after controlling for SRS, which suggested that SRS introduces a spurious correlation between J-REF and self-consciousness. This result emphasizes the strong relationship between IoR-P and self-consciousness. On the other hand, after controlling for J-REF, the correlations of SRS with the three dimensions of schizotypy became weaker, which suggested that J-REF assesses the schizotypal aspects and introduces a spurious correlation between SRS and schizotypy. These results are similar to the assumptions of Lenzenweger et al. [13]. This study also showed that controlling J-REF and SRS makes the difference between IoR-S and IoR-P clearer. To discuss their specific aspects, in the following section, IoR-S refers to the J-REF that controls SRS, and IoR-P to SRS, which controls J-REF.

The results of the hierarchical regression analyses revealed the contributions of positive schizotypy, negative schizotypy, and disorganization of IoR-S. Positive schizotypy corresponds to positive symptoms of schizophrenia and shows the presence of symptoms that normal people usually do not experience. The contribution of disorganization indicates poor organization and expression of actions and thoughts. These results suggest the cognitive and behavioral manifestations of cognitive slippage. The negative contribution of negative schizotypy to J-REF can be interpreted as an increase in social contact causing IoR-S under controlling negative affect and social anxiety (e.g., [59]). Therefore, IoR-S may be caused by the interaction between cognitive slippage and social contact. There was no significant increase in $R^2$ when public and private self-consciousness were entered. This result supports the notion that self-consciousness may not be included in the IoR-S manifestation process.

However, self-consciousness specifically predicted the manifestation of IoR-P. As expected, public self-consciousness strongly predicted IoR-P. This means that IoR-P is caused by shifting attention to external aspects (i.e., face, appearance) in the relationship with others. Moreover, the results showed that decreased private self-consciousness predicted the manifestation of IoR-P. Private self-consciousness refers to an attentional trait wherein one tends to focus on their internal state (i.e., affect, cognition, and image) [38]. This result indicates that individuals who tend to focus on their internal state might be aware that they link others' behaviors to themselves and modify excessive linkages. This result indicates that the facilitation of reflection on one's internal state could contribute to a decrease in IoR-P. There was no significant increase in $R^2$ when the three dimensions of schizotypy were entered, although there was a positive contribution of negative schizotypy to IoR-P. This result indicates that IoR-P may not be caused by schizotypal features.

One of the meaningful features was obtained by mutually controlling for IoR-S and IoR-P; social anxiety contributed to both IoR-P and IoR-S. As mentioned above, they share the same feature, namely "to misunderstand that one is being observed by others." In addition, IoR-P contributes to public self-consciousness and positive/negative affect. Social anxiety is thought to be caused by a negative interpretation bias toward ambiguous or neutral social events [60]. It is assumed that IoR-P shares a manifestation process with social anxiety, in which enhanced public self-consciousness in social contact causes social anxiety. It is also assumed that decreased pleasure in social situations enhances the sensitivity of IoR-P. It is believed that entering social situations with negative affect leads to IoR-P. Therefore, IoR-P could have high

reactivity to negative affect in social situations, including social anxiety, and could be caused by cognitive distortion [61], which implies negative cognition in social situations.

In contrast, the contribution of social anxiety to IoR-S provided an interesting perspective, although there was no contribution of affect and self-consciousness to IoR-S. Prior studies have shown a relationship between schizotypy and social anxiety (e.g., [31, 62]). This is consistent with a previous study [26], which showed that social anxiety heightened J-REF. This result might be because IoR-S and social anxiety share the feature of misunderstanding that one is being observed by others, as mentioned above. However, hierarchical regression analyses clearly showed that the contribution of social anxiety on IoR-S was weakened by entering schizotypy; on the other hand, that on IoR-P was weakened by entering self-consciousness, whereas the contributions of social anxiety on IoR and IoR-P are almost same at step 3 ($\beta$ = .28, .31, $ps$ < .001, respectively). Hence, these results imply the existence of qualitatively different aspects of social anxiety that would cause a distinct IoR.

These findings emphasize that IoR-S and IoR-P are distinct psychological phenomena with different predictors, although social anxiety is commonly involved in their manifestation. This is supported by the fact that SRS scores are normally distributed, whereas J-REF scores are not. This study also showed that IoR-S and IoR-P are heavily overlapped, and research to compare the etiologies should recognize that social anxiety is an important confounding factor and should be properly controlled.

The social aspects of the IoR are essentially critical. For example, Bell et al. [63] argued that the theme of delusions is mainly social (e.g., [64]), and similarly, delusion-like beliefs may arise from normal adaptive social processes (e.g. [65, 66]). In other words, having delusion-like beliefs is functionally rational [67], while the content is irrational. Hence, focusing on the social aspects of IoR is quite important because it broadens our perspective not only in the pathological aspects of this phenomenon, but also to its social adjustment benefits in healthy individuals.

The results of this study share some similarities with previous studies using the REF scale. Consistent with Lenzenweger et al. [13], the present study suggests that positive schizotypy predicts IoR-S. More recently, Ceballos-Munuera, et al. [68] showed that aberrant salience has the mediating role between IoR and psychotic dimension. Aberrant salience [69] refers to the experience of normal events as remarkable, which is specific to schizotypy and Schizophrenia. The concept of aberrant salience might be a promising perspective to differentiate the etiology of IoR-S and IoR-P, and emphasize that some kind of self-disorder [18] could be involved in the manifestation of IoR-S. For example, Nelson et al. [70] assumed that aberrant salience contributed to hyper-reflexivity [71], which refers to "excessive attention being paid to aspects of experience that are normally tacit and remain in the 'background' of awareness" [70]. In differentiating the etiology of IoR-S and IoR-P, it might also be important to control aberrant salience as well as social anxiety.

Regarding the result of CFAs, this study sheds light on the nature of IoR. Similar to Rodriguez-Testal et al. [27], this study supported a model of one second-order factor and five first-order factors using the oblique rotation method and indicated a better fit than the unidimensional model, although both models' fit indices were high enough but slightly different. Contrarily, Lenzenweger et al. [13] showed the first-order factor model using varimax rotation. Thus, this suggests that the subscales of REF highly correlate with each other and that each subscale of REF could represent subtle but meaningful qualitative differences. Indeed, Lenzenweger et al. intended to cover relevant experiences, such as intrapersonal and interpersonal reflection, toward persons or animals with relatively positive, neutral, or negative valences, and so on [13]. However, a true-false scale that the original and Japanese version of REF employed have difficulties as an ordinal scale for the statistical analysis, such as factor analysis

and reliability, as shown in this study. As mentioned previously, responding to the SPQ on a five-point Likert scale makes the reliability higher than a true-false scale [58]. Hence, further exploring the proper response style would result in understanding the nature of IoR better.

It was also found that age and sex did not explain J-REF. This meant that there was little variance in J-REF scores with age. This study was limited to a sample of individuals in their 20s, similar to that of Lenzenweger et al. [13], who developed the J-REF scale. Rodríguez-Testal et al. [10] showed that the mean J-REF scores peaked around the age of 15 and continued to decline, reaching stability in the 20s. The results of the present study were consistent with these findings. Rodríguez-Testal et al. [10] showed that the average J-REF score for teens was 8.32 and the average for those in their 20s was 4.28, which was similar to the average for individuals in their 20s in this study. A direct comparison study is thus needed to determine how an individual's IoR-S and IoR-P transition with age.

Considering the findings of this study, the dynamic manifestation process of IoR-S and IoR-P could be delineated: Individuals can experience IoR-S and IoR-P by enhancing social anxiety when one encounters a social situation. The aspect towards which one's attention shifts, would determine the extent of IoR-P, not IoR-S—If it is their external aspects (i.e., public self-consciousness) or their internal state (i.e., private self-consciousness), IoR-P would be increased or decreased, respectively. However, IoR-S would not be influenced by the public and private self-consciousness. Those with more experience that normal people usually do not experience (i.e., positive schizotypy) and who cannot organize their thoughts and actions (i.e., disorganization) would manifest more IoR-S. As stated previously, schizotypy is formed from the interaction of genetic factors and social learning schedules [26]. A longitudinal study is needed to clarify what kind of social events could contribute to the manifestation of IoR-S and IoR-P, when encountered by individuals with the genetic background of schizotypy.

## Limitations and direction of future research

This study makes three main contributions to the literature. First, it directly demonstrates the existence of two different types of IoR in terms of their predictors. Although it is well known that both IoR-S and IoR-P are frequently evoked during the same period of life, how they interact with each other is unknown. Further findings using J-REF will make meaningful additions to the knowledge of the mechanism of IoR. Second, we developed the J-REF and confirmed its reliability and validity. Since REF indicates the risk of schizophrenia spectrum disorders [10], J-REF could be used to examine risk in the Japanese context. It is necessary to confirm whether the cut-off point of the J-REF is 13, as Lenzenweger et al. [13] showed. It also needs to be clarified whether the J-REF and SRS have different predictive powers for the risk of schizophrenia spectrum disorders. Third, this study is significant in that it examines J-REF in the context of Asia, and shows that there may not be much difference in the frequency of IoR from other cultures. The results of the present study do not deviate much from this range, indicating the non-existence of cultural differences in the frequency of IoR-S as measured by the J-REF. Although the factor structure of schizotypy is generally stable, some studies suggest that culture influences schizotypy [72]. Whether the frequency of referential thinking remains the same across cultures or in what respect cultural differences exist requires further investigation.

This study had some limitations. First, it explored the specific predictors of two distinct types of IoR; however, it was a cross-sectional study. Thus, a longitudinal study is required. Second, the results of this study were established in a community sample, and it is not clear whether the findings of this study can be applied to patients with IoR, although it is known that people with psychiatric diagnoses have more IoR. Furthermore, the mechanisms of

manifestation of IoR in various groups, such as those with particularly high schizotypy traits (i.e., high schizotypy [73]) and those with high public self-consciousness only and low schizotypy, are also unclear. It is necessary to investigate how the predictors and confounders identified in the present study engage in the manifestation of IoR. Third, the study was limited to a community sample of people in their 20s. A systematic study with various age groups is needed to determine the predictors of IoR-P and IoR-S in other age groups. Fourth, the KMO values were suitable for factor analyses but were low. Replications of factor analysis is variable. Finally, we were unable to confirm the internal consistency in one of the five subscales ("Songs, Newspapers, Books"). If the subscales are to be used in the analysis, it is necessary to confirm whether similar factors can be replicated in a Japanese sample.

## Supporting information

**S1 Table. Descriptive statistics of the J-REF (n = 600).**
(DOCX)

**S2 Table. Descriptive statistics of the J-REF and gender differences (n = 600).**
(DOCX)

**S3 Table. The Japanese version of the Referential Thinking Scale (J-REF).**
(DOCX)

**S4 Table. Descriptive statistics of the J-REF and other scales (n = 600).**
(DOCX)

**S5 Table. Correlation of the J-REF and other scales (n = 600).**
(DOCX)

**S6 Table. Partial correlation of scales controlling for J-REF and SRS (n = 600).**
(DOCX)

## Acknowledgments

We would like to express our gratitude to Dr. Ryota Hashimoto and Dr. Sachiko Morimoto for their support in the translation process of the J-REF and to Dr. Taka-Mitsu Hashimoto for statistical assistance.

## Author Contributions

**Conceptualization:** Jun Sasaki, Seiji Muranaka, Kotomi Arahata, Atsushi Sato.

**Data curation:** Jun Sasaki, Kotomi Arahata.

**Formal analysis:** Jun Sasaki, Seiji Muranaka.

**Funding acquisition:** Jun Sasaki.

**Investigation:** Jun Sasaki, Atsushi Sato.

**Methodology:** Jun Sasaki, Seiji Muranaka, Atsushi Sato.

**Project administration:** Jun Sasaki.

**Resources:** Jun Sasaki, Kotomi Arahata.

**Software:** Seiji Muranaka.

**Supervision:** Jun Sasaki.

**Validation:** Jun Sasaki, Seiji Muranaka.

**Writing – original draft:** Jun Sasaki, Seiji Muranaka.

**Writing – review & editing:** Seiji Muranaka, Atsushi Sato.

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
