## [Decision Letter · Decision Letter 0]

12 Jan 2023

PONE-D-22-28041

Developing and validating the Japanese version of the Referential Thinking Scale: A cross-sectional study

PLOS ONE

Dear Dr. Sasaki,

Thank you for submitting your manuscript to PLOS ONE. After careful consideration, we feel that it has merit but does not fully meet PLOS ONE’s publication criteria as it currently stands. Therefore, we invite you to submit a revised version of the manuscript that addresses the points raised during the review process. The reviewers feel that the objectives and hypotheses stated in the manuscript need further clarification and organization, e.g. the psychometric objectives are not clearly specified. The reviewers also note that the methodology and statistical analyses need revisions. 

We look forward to receiving your revised manuscript.

Kind regards,

Alex Schaefer, PhD

Associate Editor

PLOS ONE

“SM received personal fees from a for-profit company CureApp Inc.”

Reviewers' comments:

Reviewer's Responses to Questions

**Comments to the Author**

1. Is the manuscript technically sound, and do the data support the conclusions?

Reviewer #1: Yes

Reviewer #2: Yes

2. Has the statistical analysis been performed appropriately and rigorously? 

Reviewer #1: Yes

Reviewer #2: No

3. Have the authors made all data underlying the findings in their manuscript fully available?

Reviewer #1: Yes

Reviewer #2: Yes

4. Is the manuscript presented in an intelligible fashion and written in standard English?

Reviewer #1: Yes

Reviewer #2: Yes

5. Review Comments to the Author

Reviewer #1: PONE-D-22-28041. Developing and validating the Japanese version of the Referential Thinking Scale: A cross-sectional study.

This is an interesting study, very well presented and developed. The authors are well aware of the fundamental aspects and importance of self-referential processing, particularly in the form of ideas of reference. Although it is in fact a psychometric study with indications of reliability and validity for the Japanese version of the REF referential thinking scale, a novel approach is also offered that differentiates ideas of reference related to paranoia (IoR-P), and related to the schizophrenia spectrum (IoR-S). It is considered a valuable work, and hopefully it will be the basis for longitudinal studies that will allow the deepening of the onset of delusional disorders, whether or not of referential type or referential content.

The introduction is well presented, highlighting the main aspects of the ideas of reference, although with a main focus on the contribution of Lenzenweger, after all, the main author in the development of the REF scale, indebted to Meehl's contributions. Although there are many contributions that could be pointed out on the ideas of reference, the approach of its authors seems to us coherent.

However, the social component of the IoR is perhaps mentioned in a very lateral way, being essential to its definition (external events and the action of others are alluded to). Moreover, the greater development or emphasis on the social characteristic would fit in with later analyses related to social anxiety. It is recommended that the text DOI: 10.1177/2167702620951553 be analyzed, because of the importance of this social aspect, and it frames IoR not only in a pathological sense, but in the context of our development as a species.

In the introduction reference is made to a classic in the study of delusions, Ernst Kretschmer, and we share the importance of this historical mention. However, ideas of reference are emphasized in normal functioning and the adolescent stage, when it may be a more important reference with respect to the origin or basis of the reference sensitive delusion, which is admittedly rather transient (asthenic character), but also the possibility of evolution to paranoia (stenic character). Perhaps it would be better to specify this mention.

Also in the introduction, the more current approach of authors such as Sass, Parnas, Henriksen, Raballo... who point out the importance of alterations of the self, at the basis of schizophrenia, and therefore, would presumably be more linked to the IoR-S, is missing. In fact, these authors incorporate the ideas of reference in instruments such as the EASE for early or prodromal detection of schizophrenia. Perhaps a mention along these lines could strengthen the discussion, given this differentiation obtained in this study. DOI: 10.1016/S2215-0366(20)30007-9 may also be reviewed.

From the methodological and statistical analysis point of view, the work is very good, correct, flawless. The instrument preparation procedure, with all the steps followed, is excellent. Very careful steps have been taken for the preparation of the instrument items, an aspect of the work that should be highlighted.

However, since they indicate a Heywood case, it is not clear how these participants are selected, was it therefore an ad-hoc decision, perhaps this should be clarified.

It also generates doubt that both KMO procedures, it is assumed that, with the initial sample of 300 participants, and later of 600, values below the recommended values are obtained, is this correct? Or perhaps it is not sufficiently clear, in view of the degrees of freedom shown. This aspect, apart from the fact that the sample is limited, should be highlighted in the limitations of the work.

Authors are strongly recommended to review the format of the tables they include, following APA standards, apart from the fact that they are displayed altered, with information in the cells occupying more than one line, etc.

The discussion is good. Perhaps too much clinging to the results, with less comparison with what it means with respect to other works. For example, given that the results have many similarities to those obtained by Rodriguez-Testal et al, what are the implications given that it is not exactly what was achieved by Lenzenweger et al?

Perhaps, and tentatively, the aforementioned authors propose a relationship of IoR with aberrant salience (doi: 10.3389/fpsyg.2022.878331). It would be interesting, based on the results obtained and the differentiation of IoR-P and IoR-S, what relationship do the authors suggest? It is clear that they are different methodologies and a different approach, but perhaps the authors can raise some idea for further studies. Is it possible that the stages described by Klaus Conrad particularly cover IoR-S?

The authors are congratulated for an interesting, novel and relevant article and, as noted in the previous paragraph, perhaps it is important to propose dynamic aspects so that this differentiation between IoR-S and IoR-P does not remain exclusively a separation of statistical adjustment in regression procedures.

Reviewer #2: The authors present a very interesting paper in which they describe the reliability and validity evidence for the Japanese version of the Referential Thinking Scale (REF) and present some analyses of variables that predict ideas of reference. Overall the paper is well written, but I am concerned about some methodological and statistical issues. Here are some comments that, in my opinion, could be used to improve the paper.

Introduction

The introduction addresses the constructs of interest and describes the association between referential thinking with psychotic and schizotypal symptoms among others. Findings found by other authors who have employed the REF scale of referential thinking are mentioned and the objectives and hypotheses are stated.

I found the presentation of the objectives and hypotheses unclear and disorganized.

The psychometric objectives are not clearly specified; it is not sufficient to state that reliability and validity are to be examined. This should be described with greater precision. In the hypotheses of the second objective, too many justifications are included as to why the hypotheses are based on the results of other researchers; this generates confusion in the reading. These hypotheses should be described more clearly.

Method

The way in which the REF scale was disseminated is not described, was it through social networks, posters, bulletin boards, etc. ....? What type of population responded to the questionnaire, were they university students, people from the general population?

How was it controlled that 150 males and 150 females responded? Were there more than 300 responses to the survey and were subjects eliminated for gender balance?

In the "measures" section put the full title of the scale with the respective citation, not the acronyms, even if you have previously specified the meaning of the acronyms.

In the description of the REF scale, provide data on the psychometric properties of the instrument as found by the authors of the scale.

Results

Before performing the factor analysis, it would have been interesting if the authors had presented descriptive analyses of the responses to the REF scale items, as well as a contrast of means by sex.

I do not understand the authors' description from line 320 to line 329. Was the initial sample composed of 600 participants and they selected 300? If this is so, why was it not described in the methodology section. If 300 participants were randomly selected, they must show that this selection does not present significant differences in sociodemographic and clinical variables with respect to the unselected sample.

Table 1 states that the n is 600 participants, wasn't it 300? Again, this shows that the description of the sample is not at all clear.

To perform the validation of an instrument they should randomly select half of the sample and perform an AFE with one half and a CFA with the other half of the sample. Cross-validation.

The descriptive results of the scales presented in Table 3 should be presented in an annex, except for those of the REF scale, which can be described in the text.

In the hierarchical regression analyses present in addition to VIF, tolerance.

Even if the VIF results do not indicate multicollinearity, why include the reference ideas in the first step of the analyses, what is the point?

Discussion

The first sentence describing the objectives of the study should be more precise, as it responds vaguely to the results that have been carried out.

This section adequately addresses what is proper for a discussion. It contrasts the results achieved with those found by other authors, presents the limitations and future lines of action and describes the contribution of the results of the work to the literature on the subject.

6. PLOS authors have the option to publish the peer review history of their article (what does this mean?). If published, this will include your full peer review and any attached files.

Reviewer #1: No

Reviewer #2: No

---

## [Author Response · Author response to Decision Letter 0]

22 Feb 2023

Please confirm the "Response to Reviewers" file. Thank you.

---

## [Decision Letter · Decision Letter 1]

8 Mar 2023

Developing and validating the Japanese version of the Referential Thinking Scale: A cross-sectional study

PONE-D-22-28041R1

Dear Dr. Sasaki,

We’re pleased to inform you that your manuscript has been judged scientifically suitable for publication and will be formally accepted for publication once it meets all outstanding technical requirements.

Kind regards,

Cristina Senin-Calderón, PhD

Guest Editor

PLOS ONE

Additional Editor Comments (optional):

Reviewers' comments:

Reviewer's Responses to Questions

**Comments to the Author**

1. If the authors have adequately addressed your comments raised in a previous round of review and you feel that this manuscript is now acceptable for publication, you may indicate that here to bypass the “Comments to the Author” section, enter your conflict of interest statement in the “Confidential to Editor” section, and submit your "Accept" recommendation.

Reviewer #1: All comments have been addressed

Reviewer #2: All comments have been addressed

2. Is the manuscript technically sound, and do the data support the conclusions?

Reviewer #1: Yes

Reviewer #2: Yes

3. Has the statistical analysis been performed appropriately and rigorously? 

Reviewer #1: Yes

Reviewer #2: Yes

4. Have the authors made all data underlying the findings in their manuscript fully available?

Reviewer #1: Yes

Reviewer #2: Yes

5. Is the manuscript presented in an intelligible fashion and written in standard English?

Reviewer #1: Yes

Reviewer #2: Yes

6. Review Comments to the Author

Reviewer #1: The authors have made an extraordinary effort in response to all suggestions. We believe that the work has been very complete, very detailed, and very well elaborated. The authors are congratulated for their contribution to the scientific panorama on the processes related to psychosis. Congratulations.

Reference number 67, line 952 of the revised manuscript should be revised, as there are hyphens that break the words of the text.

Reviewer #2: The authors have made the suggested changes and have responded accurately to my comments. I have nothing further to add. Congratulations to the authors for the manuscript.

7. PLOS authors have the option to publish the peer review history of their article (what does this mean?). If published, this will include your full peer review and any attached files.

Reviewer #1: No

Reviewer #2: No

---

## [Editor Report · Acceptance letter]

15 Mar 2023

PONE-D-22-28041R1 

Developing and validating the Japanese version of the Referential Thinking Scale: 
A cross-sectional study 

Dear Dr. Sasaki:

I'm pleased to inform you that your manuscript has been deemed suitable for publication in PLOS ONE. Congratulations! Your manuscript is now with our production department. 

Kind regards, 

on behalf of

Dr. Cristina Senin-Calderón 

Guest Editor

PLOS ONE